# Experimental warming influences species abundances in a *Drosophila* host community through direct effects on species performance rather than altered competition and parasitism

Mélanie Thierry[1,2]*, Nicholas A. Pardikes[2], Chia-Hua Lue[2], Owen T. Lewis[3], Jan Hrček[1,2]

**1** Faculty of Science, University of South Bohemia, Ceske Budejovice, Czech Republic, **2** Department of Ecology, Institute of Entomology, Biology Centre of the Czech Academy of Sciences, Ceske Budejovice, Czech Republic, **3** Department of Zoology, University of Oxford, Oxford, United Kingdom

\* m.thierry@entu.cas.cz

**Data Availability Statement:** All raw data used for this study are available from the Zenodo database: https://doi.org/10.5281/zenodo.4362189

## Abstract

Global warming is expected to have direct effects on species through their sensitivity to temperature, and also via their biotic interactions, with cascading indirect effects on species, communities, and entire ecosystems. To predict the community-level consequences of global climate change we need to understand the relative roles of both the direct and indirect effects of warming. We used a laboratory experiment to investigate how warming affects a tropical community of three species of *Drosophila* hosts interacting with two species of parasitoids over a single generation. Our experimental design allowed us to distinguish between the direct effects of temperature on host species performance, and indirect effects through altered biotic interactions (competition among hosts and parasitism by parasitoid wasps). Although experimental warming significantly decreased parasitism for all host-parasitoid pairs, the effects of parasitism and competition on host abundances and host frequencies did not vary across temperatures. Instead, effects on host relative abundances were species-specific, with one host species dominating the community at warmer temperatures, irrespective of parasitism and competition treatments. Our results show that temperature shaped a *Drosophila* host community directly through differences in species' thermal performance, and not via its influences on biotic interactions.

## Introduction

It is becoming evident that many species are declining as the climate changes [1, 2], and increasing numbers of extinctions are expected as a result in the coming decades [3]. Animals are directly impacted by warming temperatures through changes in their fecundity, mortality, metabolic rates, body growth rate, and phenology [4–7]. Species in the tropics are likely to be more sensitive to global warming because they are closer to their upper thermal limits [3, 8],

**Funding:** We acknowledge funding support of JH from the Grantová Agentura České Republiky no. 17-27184Y. MT was further supported by Jihočeská Univerzita v Českých Budějovicích (GA JU no. 038/2019/P), and OTL by the UK Natural Environment Research Council (NE/N010221/1).

**Competing interests:** The authors have declared that no competing interests exist.

and the predicted increase in temperatures by a few degrees would exceed their thermal maxima. Ectotherms, such as insects, have particularly narrow thermal limits and are facing severe declines in abundances with rising temperature [9]. Warming temperatures directly affect physiology and demography depending on species' thermal tolerances (i.e., their ability to survive exposure to extreme temperatures) and their thermal performance (i.e., their fitness-related traits over a range of temperatures). Both thermal tolerance and thermal performance are expected to influence population sizes and community structure with ongoing global warming [5].

However, ecological communities are not defined solely by the species that compose them, but also by the way those species interact with one another, via both trophic and non-trophic interactions [10, 11]. Trophic interactions, such as predation, herbivory, or parasitism have strong effects on community composition and evenness [12, 13]. Non-trophic interactions such as competition and pollination are also ubiquitous and can alter community composition in many ways (e.g. if some species are competitively excluded, or if species coexistence is enhanced) [14–16]. Trophic and non-trophic interactions act together to structure ecological communities [17–19], and a theoretical understanding is emerging of how these different types of interactions shape the structure and dynamics of more complex ecological networks [20]. However, empirical evidence on the combined effects of trophic and non-trophic interactions on the structure of terrestrial species-rich communities remain sparse. Moreover, global warming may modify such mechanisms structuring ecological communities, since warming temperatures are expected to have direct effects on both component species and their interactions [21, 22]. Temperature can alter resource-consumer interactions via its effects on metabolic processes such as growth and reproduction, and change in behaviors [23–25]. The main mechanisms behind species interactions response to climate change are the differences in effects among interacting species, such as asymmetrical responses in their phenology [26], growth rate [27], and body mass [28]. Furthermore, changes in the outcome of species interaction with warming temperatures can have cascading effects on individual fitness, populations and communities [25, 29, 30]. Despite calls for more investigations of how species interactions respond to global climate change [31, 32], most such studies focus either on aquatic systems [21, 33], on a single interaction type [34], or on a small number of species [35]. We urgently need more data to predict how environmental changes modify different types of interactions (both trophic and non-trophic) in more complex ecological networks [36, 37].

Insect host-parasitoid communities are excellent model systems to investigate how species and their interactions respond to warming temperatures [14]. Parasitoids are insects which develop in or on the bodies of arthropod hosts, killing the host as they mature, and playing an important role in regulating host populations in both natural and agricultural ecosystems [38]. As ectotherms, many parasitoid traits involved in species interactions are sensitive to changes in temperature [39, 40]. Empirical studies suggest that global warming could weaken top-down control by parasitoids by increasing parasitoid mortality, by decreasing parasitoid virulence and/or increasing host immune response, and by increasing host-parasitoid asynchrony, thus increasing the frequency of pest outbreaks [41–43]. However, most studies of host-parasitoid interactions are limited to a pair of interacting species, and it is unclear how host-parasitoid communities respond to warming temperatures when more complex systems are considered [14, 44]. Community level responses to global warming may depend on how species interact, and the effect of species interactions on community structure might change depending on environmental conditions. For instance, parasitoids can mediate host coexistence, but the outcome may depend on temperature [45]. Furthermore, competitive interactions among hosts can affect the responses of species and communities to environmental changes [30], but such responses may differ for intraspecific and interspecific competition

[46]. Thus, to help forecast the impacts of global warming on host-parasitoid communities, it will be critical to examine the combined responses of species and their interactions under simulated warming conditions [47].

In this study, we use a laboratory experiment with intra vs. inter specific competition between hosts and parasitism in a fully factorial design to investigate how temperature affects host communities directly through difference in species responses, and indirectly through effects on parasitism and competition with other host species. We used host abundances and their relative frequencies to describe the host community. We also measured host body mass as a proxy for host fitness under the different treatments, and because an increase in temperature generally produces smaller individuals, which could influence the outcome of competition [28]. We focus on a set of three *Drosophila* species which are members of a natural *Drosophila*-parasitoid community in Australian tropical rainforests [48]. We test the predictions that elevated temperature will affect the relative abundance of the hosts directly through the thermal performance of individual species, and indirectly through effects on their interactions with other species. Elevated temperatures could alter the competitive abilities of the hosts (linked to species' thermal performance) and the extent to which they are parasitized (linked to effects of temperature on parasitoid attack rates and virulence) [39], with consequences for the relative abundance of hosts in the community [14]. An interactive effect of trophic and non-trophic interactions on host relative abundances is expected due to a trade-off between resistance to parasitoids and larval competitive abilities [49]. This study aims to disentangle the direct and indirect effects of warming on structuring our focal tropical *Drosophila* community, and provides an important step forward in our understanding of the potential mechanisms driving tropical insect community responses to global warming.

## Materials and methods

### Study system

The experiment was established from cultures of *Drosophila* species and their associated parasitoids collected from two tropical rainforest locations in North Queensland, Australia: Paluma (S18˚ 59.031' E146˚ 14.096') and Kirrama Ranges (S18˚ 12.134' E145˚ 53.102') (<100 m above sea level). *Drosophila* and parasitoid cultures were established from 2017 to 2018, identified using both morphology and DNA barcoding, and shipped to the Czech Republic under permit no. PWS2016-AU-002018 from Australian Government, Department of the Environment. Three host species (*Drosophila birchii*, *D. pseudoananassae* and *D. sulfurigaster*, together accounting for ~ 48% of the host abundances sampled at the study sites [48]) and two of their natural larval parasitoid species *Asobara* sp.1 (Hymenoptera: Braconidae; Smithsonian National Museum of Natural History (NMNH) reference vouchers USNMENT01557096 [BOLD sequence accession: DROP042-21] and USNMENT01557097 [BOLD sequence accession: DROP043-21] and *Leptopilina* sp.1 (Hymenoptera: Figitidae; NMNH reference vouchers USNMENT01557104 [BOLD sequence accession: DROP050-21] and USNMENT01557117 [BOLD sequence accession: DROP053-21]) able to parasitize all three host species were used in this experiment. The parasitoid species are new undescribed species unambiguously identified by the above vouchers and sequences in order for this paper to be linked to them once they will be formally described. Data on thermal performance of the three host species have been previously measured by MacLean, Overgaard, and collaborators [50, 51] (Table 1). All cultures were maintained at 23˚C on a 12:12 hour light and dark cycle at Biology Centre, Czech Academy of Sciences. *Drosophila* isofemale lines were maintained on standard *Drosophila* medium (corn flour, yeast, sugar, agar and methyl-4-hydroxybenzoate) for approximately 15 to 30 non-overlapping generations. To ensure genetic variation, five lines from each host

**Table 1. Host species thermal tolerance upper limit ($CT_{max}$) and thermal performances: Optimal temperature ($T_{opt}$) and thermal breadth ($T_{breath}$ defined here as the range where performance is above 80% of optimal) for overall species fitness (product of fecundity, developmental success and developmental speed) and fecundity measured as egg-laying capacity ± SD.** Data are from [51].

| Host species | *D. birchii* | *D. pseudoananassae* | *D. sulfurigaster* |
|---|---|---|---|
| $CT_{max}$ | 38.51 ± 0.32 | 39.02 ± 0.32 | 36.55 ± 0.11 |
| Fitness $T_{opt}$ | 25.33 ± 1.05 | 24.00 ± 0.45 | 24.72 ± 0.73 |
| Fitness $T_{breath}$ | 4.27 ± 0.57 | 5.15 ± 0.36 | 4.51 ± 0.31 |
| Fecundity $T_{opt}$ | 26.18 ± 0.62 | 24.62 ± 1.52 | 24.84 ± 0.72 |
| Fecundity $T_{breath}$ | 5.37 ± 1.16 | 9.31 ± 1.11 | 5.26 ± 0.44 |

species were combined to establish mass-bred lines immediately before the start of the experiment. Isofemale lines of parasitoid lines were maintained for approximately 10 to 20 non-overlapping generations prior to the start of the experiment by providing them every week with 2-day-old larvae of *Drosophila melanogaster*. This host species is not present naturally at the field locations where hosts and parasitoids originated, and was not used in the experiment, thus avoiding bias of host preferences. Single parasitoid isofemale lines were used.

## Experimental design

To disentangle the effects of warming temperatures on host species and their interactions, we manipulated the presence of parasitoids and interspecific competition between host species in a fully factorial design (Fig 1) at ambient and elevated temperatures. We aimed to study the independent and combined effects of parasitism and host competition when both forms of antagonistic interaction occur at strong (but realistic) levels. As the focus of the experiment was to compare the direct and indirect effects of warming temperatures on host communities, competitive interactions between parasitoids were not assessed nor manipulated, but potentially present in all treatments with parasitoids. Parasitoid preferences were not quantified, but the two parasitoid species used were able to parasitize all three hosts species during trials.

Transparent plastic boxes (47cm x 30cm x 27.5cm) with three ventilation holes (15 cm in diameter) covered with insect-proof nylon mesh served as the experimental units (S1 Fig). Each box contained three 90 mm high and 28 mm diameter glass vials containing 2.5 mL of *Drosophila* food medium. Interactions were manipulated by establishing vials containing a single host (Fig 1A and 1C) or multiple host species (Fig 1B and 1D), and by including (Fig 1C and 1D) or excluding (Fig 1A and 1B) parasitoids. A total of 60 three-day-old virgin adult hosts, with 1:1 sex ratio, were placed in each vial to allow mating and oviposition (i.e., a total of 180 adults per box) and removed after 48 hours. In the multi-host treatment, the 60 hosts were split evenly across the three species (i.e., 20 adults for each species). The density of adult hosts was selected based on preliminary observations to achieve a high level of resource competition (i.e., the density at which strong intraspecific competition was observed for all host species; S1 Table) while keeping the number of adults for each of the three host species and the total number of adult hosts consistent across treatments and species. The treatment allowed competition both at the adult stage for oviposition space, and at the larval stage of their offspring for food resources [52, 53], but we did not aim to identify which was the primary source of competition. All results relate to the host offspring (their abundances and frequencies).

For treatments that included parasitoids (Fig 1C and 1D), ten parasitoids (3–7 days old, 1:1 sex ratio) from each species (n = 2, i.e., 20 parasitoids per box), corresponding to 9% of the total number of adult hosts, were placed in a box immediately after the hosts were removed (at 48h) and remained in each box for 72 hours, creating high but realistic parasitoid pressure (within the range of parasitism rate observed in this system in nature: 8–42% [48]). Vials were

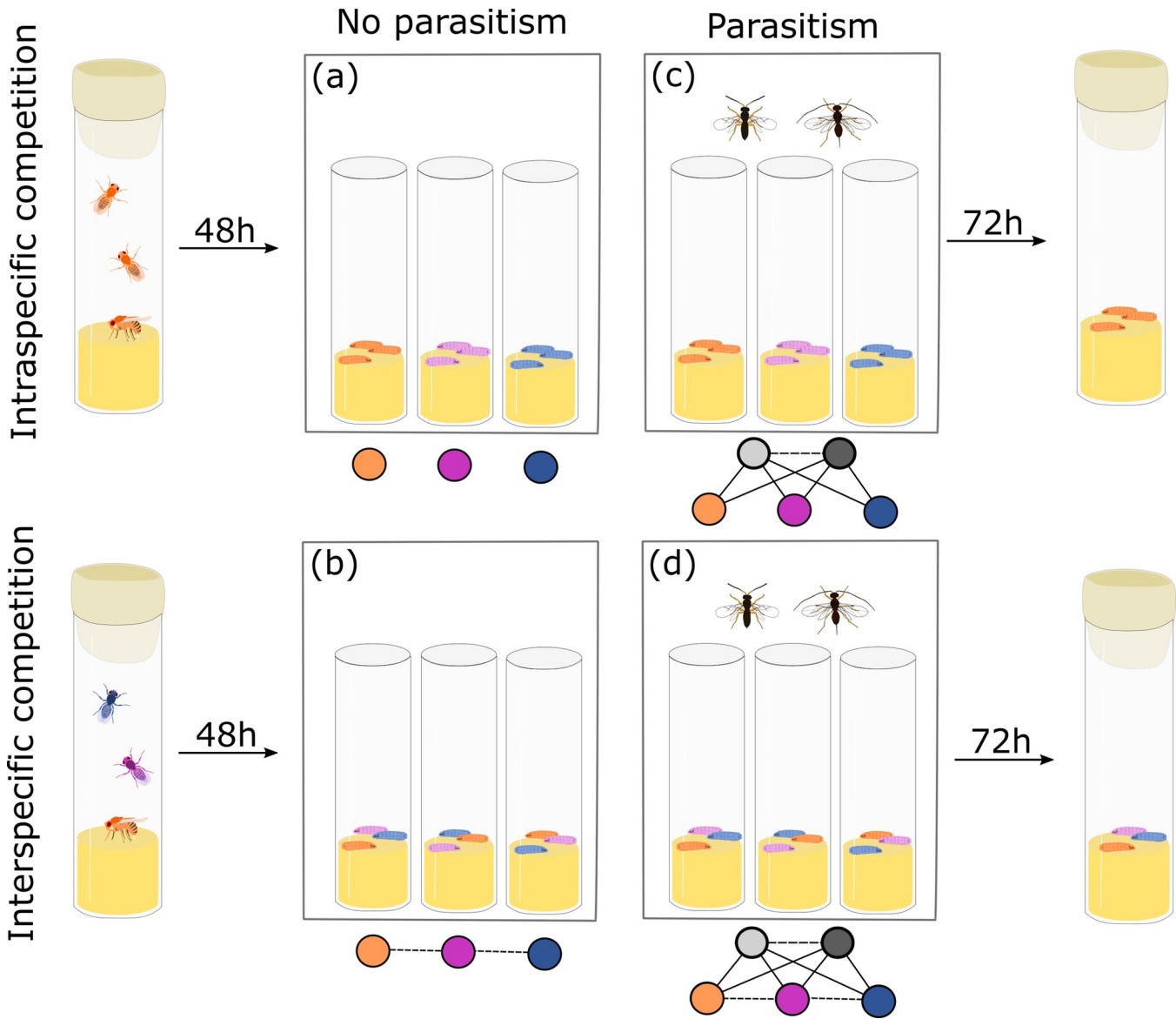

**Fig 1. Schematic representation of the steps of the protocol and the experimental treatments.** Orange, pink, and blue nodes represent the three host species, and white and grey nodes represent the two parasitoid species. Solid arrows show possible trophic interactions, and dashed arrows show possible competitive interactions in each treatment. The type of competition between host species (intraspecific/interspecific) and presence or absence of parasitoids in the cages were manipulated in a fully factorial design: a) intraspecific competition, b) interspecific competition, c) intraspecific competition with parasitism, and d) all interactions.

removed from the boxes simultaneously with the parasitoids (72 hours after parasitoid introduction), and individually sealed. Each treatment combination was replicated once across four time-blocks, and each treatment and replicate were therefore represented by three vials. The duration of the experiment corresponded to a single generation of both the hosts and the parasitoids (i.e., about 30 days for the species with the longest developmental time to emerge).

The experimental temperatures were chosen to simulate current mean yearly temperature at the two study sites [48]: 23.2 ± 0.4°C (65.9 ± 2.8% humidity), and projected temperatures representing a plausible future scenario under climate change: 26.7 ± 1.0°C (65.1 ± 2.8% humidity). The simulated difference was therefore 3.5°C (projected change in global mean

surface temperature for the late 21$^{st}$ century is 3.7˚C for the IPCC RCP8.5 baseline scenario [54]). Vials were placed at their corresponding temperature treatment from the first day the adult hosts were introduced for mating and oviposition to the last emergence (up to 40 days). All four blocks included both ambient and warming temperature treatments.

To calculate parasitism rates for each host-parasitoid species pair, pupae from the three vials of each box were randomly sampled 12 days after the initiation of the experiment. All sampled pupae were transferred into 96-well PCR plates (on average 169 ± 30 SD pupae sampled per box) and kept at their corresponding temperature treatment until adult insects emerged (up to 40 days for the slowest-developing parasitoid species). Sampled pupae were identified to their corresponding host species on the basis of pupal morphology (S2 Fig), and the outcome was recorded as either a host, a parasitoid, an empty pupal case, or an unhatched pupa. We assumed that any pupae which were empty at the time of sampling resulted in adult hosts because this period was too short for parasitoids to complete development and emerge. We calculated parasitism rates from the pupae sampled in plates only. Parasitism rates for each host-parasitoid pair were calculated as the proportion of each parasitoid species that emerged from the total number of sampled pupae of each host species.

All hosts that emerged (from both vials and sampling plates) were used to quantify the following aspects of host community structure: abundances of each host species, and their relative frequencies (i.e., the fraction of all host individuals belonging to each host species). All hosts and parasitoids that emerged from vials before and after subsampling for parasitism rates were collected, identified, and stored in 95% ethanol until four consecutive days of no adult emergences. Individual dry body mass of hosts was measured with 1 μg accuracy using a Sartorius Cubis ™ micro-balance. Only fully-eclosed and intact individuals were included in body mass measurements.

## Statistical analysis

All vials with fewer than ten total emergences or pupae were removed from analyses of host abundances, frequencies, and parasitism rates (S2 Table, deleted observation due to *D. sulfurigaster*), as these outcomes were associated with low success during the mating process and not with experimental treatments (results with the whole dataset can be found in S3 Table). We used 3-day-old hosts and allowed them to mate and lay eggs for 48 hours. *Drosophila sulfurigaster* females generally take 4 days to mature compared to 3–4 days for *D. birchii* females and 3 days for *D. pseudoananassae*, which could explain the low abundances sometimes observed for *D. sulfurigaster* compared with the two other host species.

Data were analyzed with generalized linear models (GLMs). After testing for overdispersion of the residuals, abundance data were modeled using a negative binomial error distribution, host body mass using a gaussian error distribution, and frequencies of host species and parasitism rates using a quasibinomial error distribution. Parasitism (two levels), type of competition (two levels), host species (three levels), parasitoid species (two levels), and temperature (two levels) were included as categorical predictor variables within each model. Blocks were included in the models as a fixed effect. Each two-way interaction was tested and kept in our models if judged to be statistically significant on the basis of backward selection using Likelihood-ratio tests. Interaction between temperature and parasitism, temperature and competition, and parasitism and competition were systematically kept in our models as the experiment was designed to test for the significance of these interactions. The three-way interaction between temperature, parasitism, and competition was tested for host abundances, host frequencies, and host body mass, but was not significant. Significance of the effects was tested using type III analysis of deviance with F-tests.

Post-hoc multiple comparisons were performed using the *emmeans* package, and P-values were adjusted using the Tukey method. Model assumptions were verified with the *DHARMa* package. All analyses were performed using R 3.5.2 [55] with the packages *stats*, *MASS* [56], *car* [57], *performance* [58], *DHARMa* [59], and *emmeans* [60].

## Results

In total, 7627 individuals (7063 hosts and 564 parasitoids) were reared across all treatments and replicates (238.3 ± 13.3 SD on average per box). Across all treatments and replicates, a total of 2717 pupae were sampled to estimate parasitism rates, of which 2227 (82%) produced an adult host or parasitoid. Mean host abundances, host body mass, and parasitism rates are presented for each treatment in S4 Table. We focused on the effects of temperature, parasitism, competition and their interactions on host abundances, host frequencies, and host body mass (Table 2).

### Direct effect of warming on the host community

The effect of temperature on host relative abundances varied significantly across host species (Table 2 and Fig 2). At 23˚C, *D. birchii* and *D. pseudoananassae* had similar relative abundances across treatments (mean frequency of *D. birchii* = 0.426 ± 0.05; mean frequency of *D. pseudoananassae* = 0.471 ± 0.05 for all treatments combined at 23˚C). At 27˚C, *Drosophila pseudoananassae* relative abundances increased by 12.8% (Post Hoc odd ratio (OR) = 0.336, P < 0.0001) while *D. birchii* relative abundances decreased by 56.1% (Post Hoc OR = 3.190, P < 0.0001) (mean frequency of *D. birchii* = 0.187 ± 0.02; mean frequency of *D. pseudoananassae* = 0.743 ± 0.02 for all treatments combined at 27˚C). The change in frequency of *D. sulfurigaster* with temperature was not significant (at 23˚C: 0.178 ± 0.03, at 27˚C: 0.118 ± 0.02; Post

**Table 2. Table showing the effect of temperature (23˚C or 27˚C), parasitism (present or absent), competition between host species (intraspecific or interspecific), host species (n = 3), parasitoid species (n = 2), interactions between terms, and block (n = 4) on host abundances, host frequencies, host body mass, and parasitism rate.**

|  | Df | Host abundances | | Host frequencies | | Host body mass | | Parasitism rate | |
|---|---|---|---|---|---|---|---|---|---|
| **Temperature** | 1 | 1.41 | (ns) | 0.47 | (ns) | 1.88 | (ns) | 4.89 | * |
| **Parasitism** | 1 | 21.80 | *** | 0.03 | (ns) | 2.98 | (ns) | - | - |
| **Competition** | 1 | 0.15 | (ns) | 0.06 | (ns) | 10.76 | ** | 1.14 | (ns) |
| **Host species** | 2 | 27.07 | *** | 64.7 | *** | 426.64 | *** | 2.47 | (ns) |
| **Parasitoid species** | 1 | - | - | - | - | - | - | 2.29 | (ns) |
| **Temperature x Parasitism** | 1 | 0.26 | (ns) | 0.05 | (ns) | 0.60 | (ns) | - | - |
| **Temperature x Competition** | 1 | 0.00 | (ns) | 0.07 | (ns) | 1.32 | (ns) | 0.04 | (ns) |
| **Temperature x Host species** | 2 | 7.90 | *** | 24.12 | *** | - | - | - | - |
| **Parasitism x Competition** | 1 | 1.58 | (ns) | 0.00 | (ns) | 4.49 | * | - | - |
| **Competition x Host species** | 2 | - | - | - | - | 27.80 | *** | - | - |
| **Host x parasitoid species** | 2 | - | - | - | - | - | - | 20.23 | *** |
| **Block** | 3 | 1.02 | (ns) | 0.47 | (ns) | 4.53 | ** | 1.49 | (ns) |
|  |  |  |  |  |  |  |  | - |  |
| **Df error** |  | 68 |  | 68 |  | 65 |  | 70 |  |
| **R$^2$** |  | 0.87 |  | 0.05 |  | 0.93 |  | 0.10 |  |

Degrees of freedom (Df) for each F-ratio are given for each factor and for the error. F values are presented with the significance of the effect:

(\*\*\*) P < 0.001,

(\*\*) P < 0.01,

(\*) P < 0.05, (ns) P > 0.05.

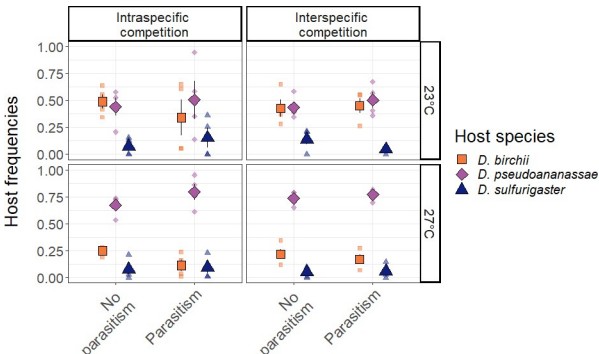

**Fig 2. Effect of experimental treatments on host frequencies.** Experimental warming changed the frequencies of hosts for all treatments. See Fig 1 for detailed description of the treatments. The small points represent the values from each block, the large points represent the grand mean, and the bars represent standard errors of the means.

Hoc OR = 1.361, P = 0.440). Elevated temperature had no effect on host body mass ($F_{1,65}$ = 1.88, P = 0.175, S3 Fig).

## Effect of biotic interactions on the host community

Parasitism significantly reduced mean abundances of all three host species by 50 ± 0.22 (SEM) hosts on average across species (β = -0.339, $F_{1,68}$ = 21.80, P < 0.0001; Fig 3A), and the negative effect of parasitism was consistent across host species (Table 2). Competition type did not significantly impact host abundances or relative host frequencies. Effects of competition on host body mass depended both on host identity ($F_{2,65}$ = 27.80, P < 0.0001), and on presence or absence of parasitoids ($F_{1,65}$ = 4.87, P = 0.038). *D. pseudoananassae* was the host species that varied the most in body mass with treatments (S3 Fig). Its body mass decreased with interspecific competition in the absence of parasitoids but increased with interspecific competition with presence of parasitoids. Changes in body mass for the other two host species were less pronounced.

## Indirect effect of warming on host community structure through parasitism and interspecific competition

Experimental warming significantly decreased parasitism rates for all host-parasitoid pairs (β = -0.29, $F_{1,70}$ = 4.89, P = 0.030, Table 2 and Fig 3B). However, the effects of parasitism and competition did not vary with temperature in affecting any of our measures of community structure (P > 0.05, Table 2).

## Discussion

Our results revealed that experimental warming directly affected *Drosophila* host community structure through differences in thermal performance among species, and decreased parasitism rates, without effects on host competition. However, warming did not impact the effect of parasitism on host community structure over the timescale investigated. The type of competition (intraspecific or interspecific) among hosts did not change host community structure.

Our results suggests that ongoing rises in global temperatures could directly alter arthropod host community structure through differences in thermal performance across species, as has been shown for communities of fish [61], plants [62], and insects [63]. Changes in host frequencies in warmer temperatures was primarily due to a dramatic increase in the relative abundance of a single host species, *D. pseudoananassae*, the species with the largest thermal

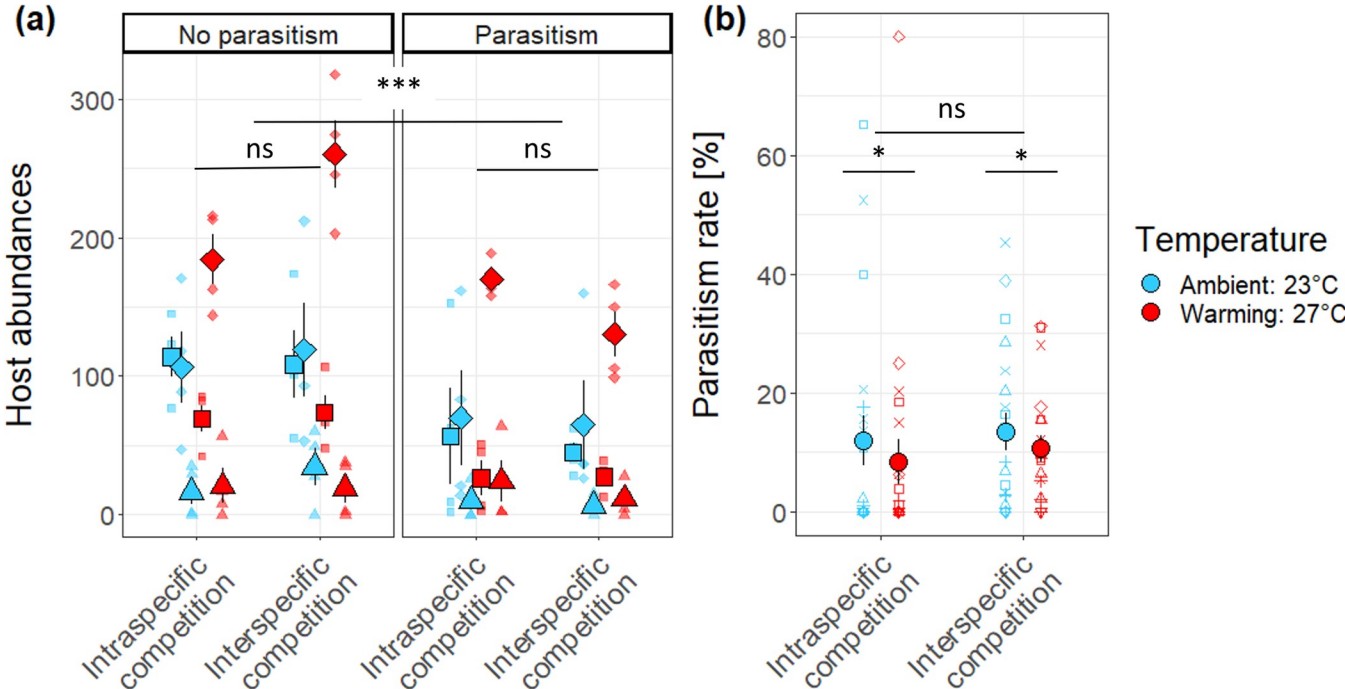

**Fig 3. Effect of experimental treatments on host community and host-parasitoid interactions.** (a) Host abundances (□: *D. birchii*, : *D. pseudoananassae*, ▲: *D. sulfurigaster*) were significantly reduced by parasitism across treatments. (b) Parasitism rates were reduced at higher temperature (□: *Asobara* sp.—*D. pseudoananassae*, ^: *Asobara* sp.—*D. birchii*, +: *Asobara* sp.—*D. sulfurigaster*, × *Leptopilina* sp.—*D. pseudoananassae*, ◇: *Leptopilina* sp.—*D. birchii*, ˇ: *Leptopilina* sp.—*D. sulfurigaster*). See Fig 1 for detailed description of the treatments. The small points represent the values from each block, the large points represent the grand mean, and the bars represent standard errors of the means. Significance of treatment effects is indicated as follows: (***) P < 0.001, (**) P < 0.01, (*) P < 0.05, (ns) P > 0.05.

performance breath [51], and our main conclusions should thus not be impacted by the low abundances sometimes observed for *D. sulfurigaster* due to mating problems. This increase occurred across all combinations of parasitism and competition treatments, and without a change in *Drosophila* body mass, suggesting a direct effect of temperature on host fecundity due to the preferred temperature of the adults for egg-laying and/or offspring egg-to-adult viability related to their thermal preference [64]. In our system, *D. pseudoananassae* distribution is limited to low elevation sites [48], and this species has a higher thermal tolerance and a bigger thermal breadth than either of the other two species considered in this study [51]. In nature, *Drosophila* species distributions are driven by differences in innate thermal tolerance limits, with low phenotypic plasticity for thermal tolerance limits in both widespread and tropical species [65]. This suggests that warming temperatures, in the context of global climate change, will have a strong effect on community composition through direct effect on fitness.

Our data also revealed a significant decrease in parasitism rates with warming. Reviews suggest that parasitism rates would decrease under global warming scenarios due to an increase in parasitoid mortality, and host-parasitoid spatial and temporal asynchrony [14, 44]. However, the presence of parasitoids significantly decreased abundances of the three host species independently of the temperature regime, suggesting that warming treatments did not decrease attack rate, but decreased successful parasitism rate [66, 67]. The decrease in parasitism rates at higher temperatures could also result from improved host immune response, decreasing the vulnerability of hosts to parasitoid attacks [68]. Therefore, host immune function responses to temperature should be considered alongside host thermal performance and tolerance to predict the effects of increasing temperatures on host communities [14]. This experiment was

performed over a single generation, so long-term consequences of decreased parasitism rates with elevated temperatures for host-parasitoid dynamic cannot be assessed, but a decrease in parasitism rates could lead to the release of hosts from top-down control. However, in the case of a simple linear tritrophic interaction, the results of Flores-Mejia *et al.* [69] suggest that parasitoid top-down control might be less sensitive to temperature than previously thought. Nevertheless, with warming temperatures, stronger host and parasitoid genotype congruence has been observed, which could decrease parasitoid diet breadth and thus decrease parasitism rates [70]. In our experiment, the role of parasitoids in lowering insect abundance was not reduced under experimental warming. However, parasitism rates were reduced, suggesting that an indirect effect of warming temperatures on the structure of the host community, mediated by parasitoids, might emerge over multiple generations.

Our results demonstrate that differences in thermal performance across host species may be a stronger determinant of how host communities respond to warming temperatures than shifts in the strength of biotic interactions in arthropod host communities. We used high, but realistic levels of competition and parasitism that would have allowed us to detect their effects on host species relative abundances if there were any. We did not find an interactive effect of parasitism and competition treatments on host abundances and frequencies. This result is in line with results from another laboratory experiment performed on the same system [71] showing that parasitism did not significantly affect host competitive coefficients. Furthermore, the type of competition between hosts did not significantly affect total host abundance, suggesting that the amount of food included was only able to support a certain number of hosts that did not vary with the type of competition. Aspects of our results contrast with those from a field transplant experiment on two species drawn from the same Australian *Drosophila*-parasitoid community [72]. Investigating fitness of *D. birchii* and *D. bunnanda* along an elevation gradient, the authors found an interacting effect between the abiotic environment and interspecific competition. However, the field experiment excluded parasitoids, and the elevational gradient studied is likely to include variations such as humidity as well as temperature, which might influence the outcome [73]. Our results also contrast with the conclusions from a systematic review on the mechanisms underpinning natural populations response to climate [47]. They found greater support for indirect effects of climate on populations through altered species interactions than direct effects. However, this review included drought in addition to temperature in the climatic variables, and the relative importance of biotic and abiotic mechanisms varied with trophic level. Moreover, the authors brought out a bias in the published studies toward temperate ecosystems and mammals, highlighting the need for more studies investigating the mechanisms driving tropical arthropod community responses to global climate changes.

Our study serves as example of the mechanisms that can be expected to drive community responses to global warming, but general conclusions on the potential impact of warming temperature on host-parasitoid networks will require replication with different species compositions and different systems. Especially, most host-parasitoid systems are tri-trophic (plants-arthropods-parasitoids), and climate warming is likely to impact host-parasitoid networks through bottom-up effects [74]. Few such experiments have been undertaken, despite the need to better disentangle direct and indirect effects of warming temperature on species communities. Ideally, future studies will also need to investigate the longer-term dynamics of such systems. Moreover, as temperatures continue to increase, species from diverse taxa are shifting their distribution worldwide to higher latitudes and elevations [75], changing their biotic environment with novel species interactions and different community assemblages [76]. Dispersal was not permitted in this study, but is likely to mediate some of the effects of warming temperature on species and their interactions [30, 77]. Understanding the mechanisms driving community responses to warming scenarios is particularly important for tropical communities,

which face more severe impacts of climate warming than temperate communities [8], and contain most threatened species of global concern [78]. Here, we demonstrate that warming had a direct effect on our focal tropical *Drosophila* host community through differences in thermal performance, without affecting the relative strength of parasitism and competition.

## Supporting information

**S1 Fig. Transparent plastic boxes (47cm x 30cm x 27.5cm) with three ventilation holes (15 cm in diameter) covered with insect-proof nylon mesh used as experimental unit allowing parasitoids to attack one of the three experimental vials containing 2-days-old host larvae for 72h.**
(PDF)

**S2 Fig. *Drosophila birchii*, *D. pseudoananassae*, and *D. sulfurigaster* pupae photography for morphological identification.** Not in scale (photo credit to Jinlin Chen).
(PDF)

**S3 Fig. Interactive effect of competition with host species, and with presence of parasitoids on mean host body mass (squares: *D. birchii*, diamonds: *D. pseudoananassae*, triangles: *D. sulfurigaster*).** See Fig 1 for detailed description of the treatments. The small points represent the values from each block and each host-parasitoid pair, the large points represent the grand mean, and the bars represent standard errors of the means. Blue: ambient temperature (23˚C), red: warming treatment (27˚C).
(PDF)

**S1 Table. Mean number of offspring per species with 10, 30, 60, 90 or 180 adult hosts (1:1 sex ratio) in a 5 mL host-media glass vial.** Choice of host number in the main experiment was based on these preliminary data to correspond to strong competition for all host species.
(PDF)

**S2 Table. Number of observations per temperature, treatments (Intraspecific competition: No interaction between host species; Interspecific competition: Direct competition between host species; Parasitism: Intraspecific competition with parasitism; All interactions: Interspecific competition with parasitism), and host species in the whole dataset, and with the reduced dataset used for analyses (excluding observations with fewer than 10 emerging insects or pupae).**
(PDF)

**S3 Table. Table showing the effect of temperature (23˚C or 27˚C), parasitism (presence or absence), competition between host species (intraspecific or interspecific), host species (n = 3), interactions between terms, and block (n = 4), on host abundances, and host frequencies for the whole dataset (without any deleted observations due to *D. sulfurigaster*).** Degrees or freedom (Df) for each F-ratio are given for each factor and for the error. F values are presented with the significance of the effect: (***) P < 0.001, (**) P < 0.01, (*) P < 0.05, (ns) P > 0.05.
(PDF)

**S4 Table. Summary table for mean (± SD) host abundances (Host ab.), individual host body mass (Host BM), total parasitism rate (PR), and parasitism rates of each parasitoid species (*Asobara* sp. and *Leptopilina* sp.) for each temperature (23 and 27˚C), treatments (competition: Intra or inter, parasitism: Present or absent), and host species (*D. birchii*, *D. pseudoananassae*, *D. sulfurigaster*).**
(PDF)

## Acknowledgments

We thank Inga Freiberga, Anna Jandová, Martin Libra and Joel Brown for their help during the collection of the data, Petr Šmilauer for his advices on the statistical analysis, and Jinlin Chen and J. Chris D. Terry for useful comments on the results. The drawings used for Fig 1 was made by Tereza Holicová.

## Author Contributions

**Conceptualization:** Mélanie Thierry, Owen T. Lewis, Jan Hrček.

**Data curation:** Mélanie Thierry.

**Formal analysis:** Mélanie Thierry.

**Funding acquisition:** Mélanie Thierry, Owen T. Lewis, Jan Hrček.

**Investigation:** Mélanie Thierry, Nicholas A. Pardikes, Chia-Hua Lue.

**Methodology:** Mélanie Thierry.

**Resources:** Jan Hrček.

**Visualization:** Mélanie Thierry.

**Writing – original draft:** Mélanie Thierry.

**Writing – review & editing:** Mélanie Thierry, Nicholas A. Pardikes, Chia-Hua Lue, Owen T. Lewis, Jan Hrček.

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
