## [Decision Letter · Decision Letter 0]

15 Jan 2021

PONE-D-20-39886

Experimental warming influences species abundances in a Drosophila host community through direct effects on species performance rather than altered competition and parasitism

PLOS ONE

Dear Dr. THIERRY,

Thank you for submitting your manuscript to PLOS ONE. After careful consideration, we feel that it has merit but does not fully meet PLOS ONE’s publication criteria as it currently stands. Therefore, we invite you to submit a revised version of the manuscript that addresses the points raised during the review process.

Dear authors, I have received the reviewers' revision of the manuscript, and overall they considered it an essential and novel experimental contribution to the subject species interaction and climate warming. Further, the results shed light on the mechanism describing direct and indirect effects on communities' structure and function under climate variance. Reviewers' primary concern, and mine, point out difficulties to follow the Methods section raising many questions to be clarified by the authors. Besides, provide details - i.e., which institution approved and protocol number if it was the case -  about species entrance in the country.

Therefore, we invite you to submit a revised version of the manuscript that addresses the points raised during the review process by 27th February.

We look forward to receiving your revised manuscript.

Kind regards,

Lucas D. B. Faria

Academic Editor

PLOS ONE

Journal Requirements:

Additional Editor Comments:

Despite some issues in the Methods section, the manuscript can be accepted under minor revision by the authors.

Reviewers' comments:

Reviewer's Responses to Questions

**Comments to the Author**

1. Is the manuscript technically sound, and do the data support the conclusions?

Reviewer #1: Yes

Reviewer #2: Yes

2. Has the statistical analysis been performed appropriately and rigorously? 

Reviewer #1: Yes

Reviewer #2: Yes

3. Have the authors made all data underlying the findings in their manuscript fully available?

Reviewer #1: Yes

Reviewer #2: Yes

4. Is the manuscript presented in an intelligible fashion and written in standard English?

Reviewer #1: Yes

Reviewer #2: Yes

5. Review Comments to the Author

Reviewer #1: This manuscript details the results of a laboratory experiment testing whether climate warming, species interactions (intraspecific competition, interspecific competition, and parasitism), or their interaction affect 3-species tropical Drosophila communities. The manuscript is mostly well-written (aside from parts of the Methods) and is skillfully couched in terms of pertinent larger questions and the literature. The experiment appears to have been carried out using sound methods, and analyzed and presented appropriately. The results should be interesting and useful to a broad audience interested in effects of climate warming on ecological communities. My main concern is that the methods – especially relating to the experimental design – are not explained as clearly as they could be. However, this should require a relatively minor revision. I have several other concerns below, all of which are minor as well. I feel confident that upon revision, this manuscript should represent a very useful contribution to understanding of ecology and global change.

Introduction:

• Lines 56-57: clarify: evidence for what exactly is lacking?

• In the part of the paragraph following that sentence, as well as in lines 77-82, it is a little difficult to follow the flow. Transitions may help.

• Last paragraph is a confusing. Clarify what is measured and why. First sentence makes it seem like you just focus on relative abundance, but the next sentence mentions body mass without fully explaining how it fits in. Clarify predictions – that sentence is confusing. Should also be more clear about what was done, i.e. overview of the experimental design.

• Lines 95-97: seems out of place at the end of the introduction.

Methods:

• Line 103: cultures were established in both 2017 and 2018? clarify.

• Lines 105-106: clarify “together accounting for ~48% of the host communities as the study sites.” Those species comprised 48% of fly individuals across the sites? Or some other meaning?

• Lines 110-112: remember to remove this note

• Why was care taken to ensure genetic diversity in the Drosophila but not in the parasitoids?

• Line 131: 15 cm diameter holes would take up half the box – should read 15 mm?

• Line 136: I take it that the adult flies were then removed after 48 hours and before the parasitoids were added – but this should be stated to clarify.

• Lines 148-149: sentence seems out of place – could be moved to earlier in the paragraph

• Figure 1 should be altered to more accurately represent experimental design. E.g. if adult flies and parasitoids were not present at the same time, don’t show them together. Also, no larvae are shown. Perhaps just show larvae instead of adult flies.

• Lines 151-152: what does this mean – that the experiment was ended when both flies and parasitoids emerged? How long was this? But next sentence (last sentence of paragraph) seems to imply the experiment was only one day – do you mean that each block was started a day apart but then ran for many days?

• First Experimental design paragraph is confusing as methods are presented out of order. It is not easy to understand what was done. Also there is at least one important hole in the methods: how long after introducing the adult flies were the parasitoids introduced?

• Lines 163-166: clarify how this projected temperature falls among climate warming scenarios, e.g. mid-range for 2100?

• What was the experimental design of the temperature treatments, i.e. were two of the time blocks warm and two normal?

• Lines 183-185: Move this justification for looking at body mass to the introduction.

• Line 188: Since D. sulfu. represented all the “failures,” and there were many, this should be pointed out in the text, not just in the supplemental info. Relatedly, might this impact the conclusions? Maybe address this in the discussion.

• Lines 198: remove “ANOVA.” Likelihood ratio tests and the “anova()” command in R have nothing to do with ANOVA (confusingly).

• Include overview of the fly species’ thermal performances rather than just citing the papers.

Results:

• Table 1 & Table S3: Remove “ANOVA,” since you ran GLMs, not ANOVAs.

• Include line numbers in second half of ms.

• “Direct effect of warming” paragraph: also mention the third species. 4th sentence: clarify rel. abundances changed from which treatment to which treatment?

• Figures are blurry – provide high definition versions.

• Fig. 3: include the shapes/species in the legend(s), rather than in the caption.

• Last part of the final sentence in the Results sounds more like discussion.

Discussion:

• First paragraph: first sentence sounds a bit general – include study species/system. Include more in this paragraph: especially, competition.

• Towards the end of second paragraph: a bit confusing that you say D. pseudoananassae did not have the highest thermal tolerance in the paper but did in this experiment. Why mention the first part about the other study?

• The claim that climate change will affect tropical communities more severely needs to be backed up with a citation.

• Last sentence should be moved to earlier in the discussion.

References:

Need to be edited.

Reviewer #2: Thierry et al. performed a community-level experiment to understand the direct and indirect effects of warming on host species. They found that warming directly affect the abundance of host species and decrease parasitism rates for host-parasitoid pairs. Besides, the top-down control was still maintained by the presence of parasitoids decreasing host abundance regardless of the temperature. They conclude that the warming had a direct effect on host community structure through differences in host thermal performance but not via biotic interactions (competition and parasitism).

The manuscript is well written and makes novel contributions to a topic underexplored in the literature because it access the impact of warming on a community-wide system incorporating direct and indirect effects. Although the experiment was conducted over a single host generation, the results shed a light on how the direct and indirect effects could play a role in structuring a host community under warming conditions.

Detailed comments as follows:

Abstract:

line 23: Would it be more clear to use global climate change instead of global change?

line 30: it may be more clear if authors specify what metric was used when saying ‘host communities’. E.g host species abundance and frequency

Introduction: overall, really well written and provide a good context for the predictions stated at the end of this section.

Lines 88 - 91: If possible, I would rephrase the prediction for clarity. e.g We tested the predictions that the elevated temperature will affect the relative abundance of the host directly through the thermal performance of the species, and indirectly through effects on their interactions such as hosts competitive abilities and parasitoids virulence.

Throughout the manuscript, you use the terms: attack rates, parasitoid virulence, parasitism (yes or not) and parasitism rates (host-parasitoid species pair). I think the manuscript would benefit from explaining or deleting some of these terms. For example: I think it would be clearer if you could specify in the manuscript what do you consider as the parasitoid virulence metric? Is it the parasitism rates, the ‘yes or no parasitism’, both or none? In the Line 92 the word ‘parasitism’ refers to: ‘yes or no parasitism’ or to ‘parasitism rates (host-parasitoid species pair)’?. Please, could you rephrase the sentence in lines 92 – 93 (…while the effects on parasitism will be linked to effects of temperature on parasitoid attack rate and virulence.) for clarity?

Methods:

Line 116: what do you mean by revive genetic variation? Would it be to ensure or to rescue?

Lines 133 – 135: the single species and multiple species treatments were only for hosts right? The parasitoid-host species interaction was a sample from the single and multiple host species treatment, right? In these lines I would add ‘single-host species’ and ‘multiple-host species’ for clarity.

Line 160: In figure 1 legend we read ‘c)parasitism only’. However this treatment also includes host intraspecific competition according to what it is showing in Figure1, right? I would then change this label.

In Figure1: I think it would be more informative if you could add a dashed line (or indicate somehow) the intraspecific competition in letter a and c. Besides, in the mixed-species treatment do you think that intraspecific competition could also have happened together with the interspecific competition? If yes, do you think intraspecific could have been stronger than interspecific competition in these mixed-species treatments (figure 1 letter b and d)?

In Line 171 ‘the sampled pupae were identified to their corresponding host species’, how was it done? Do the Drosophila species have visually different pupal case? Was it how you link a parasitoid to a host species to calculate parasitism rates?

Line 176: I would make it more clear here that the parasitism rates was calculated for each host-parasitoid pair.

Line 182 we read ‘second generation started’, does it mean the F1 offspring? Are you counting the Parental generation as first?

Line 186: you mean fully-ecloded?

Results:

Line 213: do you mean host frequencies instead of host proportions?

In Figure 3 legend: in letter ‘b’ we read: (b) Parasitism was reduced at higher temperature. Do you mean here parasitism rates? In your manuscript I separate parasitism as ’yes or no’ and parasitism rates as ‘the host-parasitoid pair.’

In Figure 2 legend: do you refer to ‘proportion’ as the host frequency?

Discussion:

Below I apologize for not indicating the line number that I’m referring to. It happened because the number of the lines is not appearing in my pdf version.

In the third paragraph of the discussion you have: ‘However, presence of parasitoids significantly decreased... parasitoid virulence’. It would be more clear if there is a following up sentence to better explain this sentence and the decrease in virulence. For example, why parasitoids would decrease virulence with increasing temperature?

The idea that warming decreased parasitism rates, but did not impact the effect of the presence of parasitoid in decreasing host abundance, could be better explored in the discussion. For example, although parasitoid preference for host species was not quantified, Lavandero and Tylianakis et al 2013 shows that genetically similar parasitoids become more likely to attack genetically similar hosts in warmer sites. Therefore, there could be a shift in the parasitoid preference for a host species or even for a narrower subset of host species that would decrease parasitism rates (host-parasitoid pair) due to parasitoid competition under warming conditions.

In the fourth paragraph, fourth line we read: ‘…allowed us to detect their effects on host species relative abundances if they were any.’ Should the word ‘they’ be ‘there’?

In the last paragraph for the discussion we read: ‘The role of parasitoids as essential… after several generations’. What do you mean by suggesting that the indirect effects of the temperature on the host community through parasitism could be observed after several generations?

6. PLOS authors have the option to publish the peer review history of their article (what does this mean?). If published, this will include your full peer review and any attached files.

Reviewer #1: No

Reviewer #2: No

---

## [Author Response · Author response to Decision Letter 0]

23 Jan 2021

Dear Editor,

Thank you for considering our manuscript for publication in PLOS ONE and for the opportunity to submit a revised version. We are glad that the reviews were positive and found our paper to be interesting and novel. We appreciate the constructive criticisms made by the reviewers and the Academic Editor, and have made appropriate changes in response as detailed point by point below. We hope that the revision will allow the manuscript to be published in PLOS ONE.

Yours sincerely,

Mélanie Thierry, Nicholas A. Pardikes, Chia-Hua Lue, Owen T. Lewis & Jan Hrček

Academic editor’s comments:

Dear authors, I have received the reviewers' revision of the manuscript, and overall they considered it an essential and novel experimental contribution to the subject species interaction and climate warming. Further, the results shed light on the mechanism describing direct and indirect effects on communities' structure and function under climate variance. Reviewers' primary concern, and mine, point out difficulties to follow the Methods section raising many questions to be clarified by the authors. Besides, provide details - i.e., which institution approved and protocol number if it was the case - about species entrance in the country.

* We appreciate the positive comments on our manuscript. We have rewritten the Methods section and modify Figure 1 to clarify the experimental protocol, and have added details on the permit [line 111 in the revised manuscript].

We have added that the species were “shipped to the Czech Republic under permit no. PWS2016-AU-002018 from Australian Government, Department of the Environment” [lines 111-112 in the revised manuscript].

Additional Editor Comments:

Despite some issues in the Methods section, the manuscript can be accepted under minor revision by the authors.

Reviewers' comments:

Reviewer #1: 

This manuscript details the results of a laboratory experiment testing whether climate warming, species interactions (intraspecific competition, interspecific competition, and parasitism), or their interaction affect 3-species tropical Drosophila communities. The manuscript is mostly well-written (aside from parts of the Methods) and is skillfully couched in terms of pertinent larger questions and the literature. The experiment appears to have been carried out using sound methods, and analyzed and presented appropriately. The results should be interesting and useful to a broad audience interested in effects of climate warming on ecological communities. My main concern is that the methods – especially relating to the experimental design – are not explained as clearly as they could be. However, this should require a relatively minor revision. I have several other concerns below, all of which are minor as well. I feel confident that upon revision, this manuscript should represent a very useful contribution to understanding of ecology and global change.

* We appreciate the reviewer’s interest in our manuscript. As stated above, we have rewritten the Methods section and edited Figure 1 to clarify the experimental design.

Introduction:

• Lines 56-57: clarify: evidence for what exactly is lacking?

* We have clarified the sentence [lines 56-58 in the revised manuscript].

• In the part of the paragraph following that sentence, as well as in lines 77-82, it is a little difficult to follow the flow. Transitions may help.

* We added transitions to help the flow of the paragraphs [lines 57 and 79-82 in the revised manuscript].

• Last paragraph is a confusing. Clarify what is measured and why. First sentence makes it seem like you just focus on relative abundance, but the next sentence mentions body mass without fully explaining how it fits in. Clarify predictions – that sentence is confusing. Should also be more clear about what was done, i.e. overview of the experimental design.

* We clarified in this paragraph what was measured (“host abundances and their relative frequencies”), why (“to describe the host community”), our predictions [lines 94-99 in the revised manuscript], and an outline of the experimental design (“a laboratory experiment with intra vs. inter specific competition between hosts and parasitism in a fully factorial design”).

• Lines 95-97: seems out of place at the end of the introduction.

* We rewrite this sentence [lines 101-104 in the revised manuscript].

Methods:

• Line 103: cultures were established in both 2017 and 2018? clarify.

* We changed it to “from 2017 to 2018” for clarity [line 110 in the revised manuscript].

• Lines 105-106: clarify “together accounting for ~48% of the host communities as the study sites.” Those species comprised 48% of fly individuals across the sites? Or some other meaning?

* We specified that the number refers to host abundances sampled [line 113 in the revised manuscript].

• Lines 110-112: remember to remove this note

* We have provided a final version of the species identity text [lines 114-119 in the revised manuscript].

• Why was care taken to ensure genetic diversity in the Drosophila but not in the parasitoids?

* Including wide genetic diversity of the hosts exceeds standard practice for ecological laboratory experiments, but was not possible for parasitoids: Unfortunately, we did not have enough isofemale lines of parasitoids in culture to combine them in the same way that we did for the Drosophila. 

• Line 131: 15 cm diameter holes would take up half the box – should read 15 mm?

* 15 cm is the correct size, which takes about 1/3 of 3 of the sides. We have provided a picture of the experimental boxes in Supplementary Figure S1 for visualization.

• Line 136: I take it that the adult flies were then removed after 48 hours and before the parasitoids were added – but this should be stated to clarify.

* Yes, this is correct. We have now stated this more clearly in the manuscript and added the steps of the protocol in Figure 1.

• Lines 148-149: sentence seems out of place – could be moved to earlier in the paragraph

* We moved it to the beginning of the experimental design [lines 141-143 in the revised manuscript].

• Figure 1 should be altered to more accurately represent experimental design. E.g. if adult flies and parasitoids were not present at the same time, don’t show them together. Also, no larvae are shown. Perhaps just show larvae instead of adult flies.

* The figure has been edited as suggested.

• Lines 151-152: what does this mean – that the experiment was ended when both flies and parasitoids emerged? How long was this? But next sentence (last sentence of paragraph) seems to imply the experiment was only one day – do you mean that each block was started a day apart but then ran for many days?

* Yes, each block started a day apart and ran until everything emerged (about 30 days maximum for the species with the longest developmental time). We deleted the last sentence of the paragraph to avoid this confusion.

• First Experimental design paragraph is confusing as methods are presented out of order. It is not easy to understand what was done. Also there is at least one important hole in the methods: how long after introducing the adult flies were the parasitoids introduced?

* We added the time when the parasitoids were introduced (48h after the hosts, when we removed the adult hosts) [lines 165-166 in the revised manuscript]. We also modified Figure 1 to show the different steps of the protocol.

• Lines 163-166: clarify how this projected temperature falls among climate warming scenarios, e.g. mid-range for 2100?

* We added the projected increase in temperature for the late 21st century according to the IPCC RCP8.5 baseline scenario [lines 184-185 in the revised manuscript].

• What was the experimental design of the temperature treatments, i.e. were two of the time blocks warm and two normal?

* We specified that all four blocks included both temperature treatments [line 187 in the revised manuscript].

• Lines 183-185: Move this justification for looking at body mass to the introduction.

* We have moved it to the Introduction as suggested [lines 91-93 in the revised manuscript].

• Line 188: Since D. sulfu. represented all the “failures,” and there were many, this should be pointed out in the text, not just in the supplemental info. Relatedly, might this impact the conclusions? Maybe address this in the discussion.

* We added a discussion of this issue in the Methods section [lines 210-214 in the revised manuscript] and mentioned it in the Discussion section [lines 294-296 in the revised manuscript].

• Lines 198: remove “ANOVA.” Likelihood ratio tests and the “anova()” command in R have nothing to do with ANOVA (confusingly).

* Removed

• Include overview of the fly species’ thermal performances rather than just citing the papers.

* We have added in a new Table 1 containing host species thermal tolerance upper limit and thermal performances (optimal temperature and thermal breadth for overall species fitness and fecundity) measured by MacLean, Overgaard, and collaborators.

Results:

• Table 1 & Table S3: Remove “ANOVA,” since you ran GLMs, not ANOVAs.

* Removed

• Include line numbers in second half of ms.

* We have added the lines numbers for the whole manuscript. 

• “Direct effect of warming” paragraph: also mention the third species. 4th sentence: clarify rel. abundances changed from which treatment to which treatment?

* We added the third species in the text [lines 250-251 in the revised manuscript], and stated that the relative abundances were for all four treatments combined at 23°C versus 27°C.

• Figures are blurry – provide high definition versions.

* All figures have been updated with a higher resolution.

• Fig. 3: include the shapes/species in the legend(s), rather than in the caption.

* The shapes of panel (a) represent host species whereas the shapes of panel (b) represent each host-parasitoid pair. The figure would be too small to add a legend for each panel. Therefore, we considered that adding the legend for the shapes in the figure rather than in the caption would be confusing.

• Last part of the final sentence in the Results sounds more like discussion.

* This sentence has been removed.

Discussion:

• First paragraph: first sentence sounds a bit general – include study species/system. Include more in this paragraph: especially, competition.

* We completed the first paragraph with adding that results are on Drosophila host community, that experimental warming had no effects on host competition, and that the type of competition (intraspecific or interspecific) among hosts did not change host community structure [lines 285-289 in the revised manuscript]. 

• Towards the end of second paragraph: a bit confusing that you say D. pseudoananassae did not have the highest thermal tolerance in the paper but did in this experiment. Why mention the first part about the other study?

* In our study, we did not measure host species thermal tolerance but only recorded their abundances and frequencies at ambient and elevated temperatures. We deleted the part mentioning thermal performance optimum to avoid confusion in the Discussion.

• The claim that climate change will affect tropical communities more severely needs to be backed up with a citation.

* We have added the reference [line 359 in the revised manuscript].

• Last sentence should be moved to earlier in the discussion.

* The last sentence was moved as suggested [lines 320-323 in the revised manuscript].

References: Need to be edited.

* References have been edited through our reference manager Mendeley.

Reviewer #2: 

Thierry et al. performed a community-level experiment to understand the direct and indirect effects of warming on host species. They found that warming directly affect the abundance of host species and decrease parasitism rates for host-parasitoid pairs. Besides, the top-down control was still maintained by the presence of parasitoids decreasing host abundance regardless of the temperature. They conclude that the warming had a direct effect on host community structure through differences in host thermal performance but not via biotic interactions (competition and parasitism).

The manuscript is well written and makes novel contributions to a topic underexplored in the literature because it access the impact of warming on a community-wide system incorporating direct and indirect effects. Although the experiment was conducted over a single host generation, the results shed a light on how the direct and indirect effects could play a role in structuring a host community under warming conditions.

Detailed comments as follows:

Abstract:

line 23: Would it be more clear to use global climate change instead of global change?

* We have changed it as suggested [line 23 in the revised manuscript].

line 30: it may be more clear if authors specify what metric was used when saying ‘host communities’. E.g host species abundance and frequency

* We have specified that effects were measured on host abundances and frequencies [line 30 in the revised manuscript].

Introduction: overall, really well written and provide a good context for the predictions stated at the end of this section.

Lines 88 - 91: If possible, I would rephrase the prediction for clarity. e.g We tested the predictions that the elevated temperature will affect the relative abundance of the host directly through the thermal performance of the species, and indirectly through effects on their interactions such as hosts competitive abilities and parasitoids virulence.

* The prediction was rephrased following this suggestion [lines 94-96 in the revised manuscript].

Throughout the manuscript, you use the terms: attack rates, parasitoid virulence, parasitism (yes or not) and parasitism rates (host-parasitoid species pair). I think the manuscript would benefit from explaining or deleting some of these terms. For example: I think it would be clearer if you could specify in the manuscript what do you consider as the parasitoid virulence metric? Is it the parasitism rates, the ‘yes or no parasitism’, both or none? 

* Parasitism rate reflects both attack rate and successful parasitism rate, and relate to parasitoid virulence. We only measured parasitism rates, but we could hypothesize that the decrease in parasitism rates with warming temperature was due to a decrease in successful parasitism rate and not attack rate by looking at the effect of parasitism treatment (presence or absence) on host abundances at both temperatures. We have now made sure that the terms used were clear and consistent through the manuscript.

In the Line 92 the word ‘parasitism’ refers to: ‘yes or no parasitism’ or to ‘parasitism rates (host-parasitoid species pair)’?. Please, could you rephrase the sentence in lines 92 – 93 (…while the effects on parasitism will be linked to effects of temperature on parasitoid attack rate and virulence.) for clarity?

* The sentence was rewritten for clarity [lines 96-99 in the revised manuscript].

Methods:

Line 116: what do you mean by revive genetic variation? Would it be to ensure or to rescue?

* We have changed revive to ensure as suggested [line 126 in the revised manuscript].

Lines 133 – 135: the single species and multiple species treatments were only for hosts right? The parasitoid-host species interaction was a sample from the single and multiple host species treatment, right? In these lines I would add ‘single-host species’ and ‘multiple-host species’ for clarity.

* We have modified the sentence as suggested [lines 151-152 in the revised manuscript].

Line 160: In figure 1 legend we read ‘c)parasitism only’. However this treatment also includes host intraspecific competition according to what it is showing in Figure1, right? I would then change this label.

* We have changed the legend according to suggestion.

In Figure1: I think it would be more informative if you could add a dashed line (or indicate somehow) the intraspecific competition in letter a and c. 

* We did not add a dashed line for intraspecific competition but changed the legend of panel (c) to add intraspecific competition as suggested above. 

Besides, in the mixed-species treatment do you think that intraspecific competition could also have happened together with the interspecific competition? If yes, do you think intraspecific could have been stronger than interspecific competition in these mixed-species treatments (figure 1 letter b and d)?

* We have added a discussion about the absence of effect of the type of competition between hosts on host abundances in the Discussion section [lines 331-333 in the revised manuscript].

In Line 171 ‘the sampled pupae were identified to their corresponding host species’, how was it done? Do the Drosophila species have visually different pupal case? Was it how you link a parasitoid to a host species to calculate parasitism rates?

* Yes, we chose host species that could be tell apart from their pupae. We have now specified that pupae were identify morphologically [lines 192-193 in the revised manuscript], and have added a new Supplementary Figure S3 showing photography of the pupae for the three host species.

Line 176: I would make it more clear here that the parasitism rates was calculated for each host-parasitoid pair.

* We have clarified this [lines196-197 in the revised manuscript].

Line 182 we read ‘second generation started’, does it mean the F1 offspring? Are you counting the Parental generation as first?

* We have removed this part to avoid confusion and replace it by “until four consecutive days of no adult emergences” [line 203 in the revised manuscript]. We previously considered F0 as the parental generation and F1 as the offspring.

Line 186: you mean fully-ecloded?

* We believe “eclosed” is the correct spelling.

Results:

Line 213: do you mean host frequencies instead of host proportions?

* Yes, we have modified this according to the suggestion [line 237 in the revised manuscript].

In Figure 3 legend: in letter ‘b’ we read: (b) Parasitism was reduced at higher temperature. Do you mean here parasitism rates? In your manuscript I separate parasitism as ’yes or no’ and parasitism rates as ‘the host-parasitoid pair.’

* Yes, we have changed the legend as suggested (“parasitism rates” instead of “parasitism”) and have made sure that the terms were clear throughout the revised manuscript. 

In Figure 2 legend: do you refer to ‘proportion’ as the host frequency?

* Yes, we have changed the legend as suggested.

Discussion:

Below I apologize for not indicating the line number that I’m referring to. It happened because the number of the lines is not appearing in my pdf version.

* Apologies. Line numbers have now been added for the whole manuscript.

In the third paragraph of the discussion you have: ‘However, presence of parasitoids significantly decreased... parasitoid virulence’. It would be more clear if there is a following up sentence to better explain this sentence and the decrease in virulence. For example, why parasitoids would decrease virulence with increasing temperature?

* We have changed “parasitoid virulence” for “successful parasitism rates” [lines 309-310 in the revised manuscript], and reorganized parts of this paragraph for clarity. 

The idea that warming decreased parasitism rates, but did not impact the effect of the presence of parasitoid in decreasing host abundance, could be better explored in the discussion. For example, although parasitoid preference for host species was not quantified, Lavandero and Tylianakis et al 2013 shows that genetically similar parasitoids become more likely to attack genetically similar hosts in warmer sites. Therefore, there could be a shift in the parasitoid preference for a host species or even for a narrower subset of host species that would decrease parasitism rates (host-parasitoid pair) due to parasitoid competition under warming conditions.

* Thank you for this suggestion. We have reorganized the discussion of this result and now discuss the suggested reference [lines 318-320 in the revised manuscript].

In the fourth paragraph, fourth line we read: ‘…allowed us to detect their effects on host species relative abundances if they were any.’ Should the word ‘they’ be ‘there’?

* Yes, corrected

In the last paragraph for the discussion we read: ‘The role of parasitoids as essential… after several generations’. What do you mean by suggesting that the indirect effects of the temperature on the host community through parasitism could be observed after several generations?

* We have moved this part of the Discussion to a previous paragraph where we explained the idea in more detail [lines 320-323 in the revised manuscript].

---

## [Editor Report · Decision Letter 1]

27 Jan 2021

PONE-D-20-39886R1

Experimental warming influences species abundances in a Drosophila host community through direct effects on species performance rather than altered competition and parasitism

PLOS ONE

Dear Dr. THIERRY,

Thank you for submitting your manuscript to PLOS ONE. After careful consideration, we feel that it has merit but does not fully meet PLOS ONE’s publication criteria as it currently stands. Therefore, we invite you to submit a revised version of the manuscript that addresses the points raised during the review process.

We look forward to receiving your revised manuscript.

Kind regards,

Lucas D. B. Faria

Academic Editor

PLOS ONE

Additional Editor Comments (if provided):

Dear THIERRY, M. and authors

I have gone through the revised manuscript. Comments raised by the reviewers were responded by the authors ok. However, I still have doubts about figures (2, 3 and S7) and statistical tests.

Line 227 points out the existence of Tukey's post hoc test, however, I could not find through the text or at the figures any reference to it. Please confirm it and add the information or remove it from the text.

Figure 2, 3 and S7 labels ascribe (The small points represent the values from each block and each host-parasitoid pair, the large points represent the grand mean, and the bars represent standard errors of the means). However, I could no find a difference between point sizes. Please confirm it and add the information or remove it from the text.

Figure 3(b) label legends species as follow (□: Asobara sp. - D. pseudoananassae, ˄: Asobara sp. - D. birchii, +: Asobara sp. - D. sulfurigaster, × Leptopilina sp. - D. pseudoananassae, ◊: Leptopilina sp. - D. birchii, ˅: Leptopilina sp. - D. sulfurigaster). However, figure did not shows it. Please add it to the figure.

Tables 2 and S5 labels are quite similar (difference consist only "...for the whole dataset". Please explain the difference between them - by means of the whole dataset - in the S5 label so readers could follow the authors' idea easily. Further and regarding to F-values legend [F-values are presented with the significance of the effect: (***) P < 0.001, (**) P < 0.01, (*) P < 0.05, (.) P < 0.1, (ns) P > 0.05"] - I understand that p-values greater than 0.05 are non-significative, so "(.) P < 0.1" does not make sense to me, and I am asking to avoid it, even though R model summary gives it. Then change it at the tables (.) to (ns).

Kind regards

---

## [Author Response · Author response to Decision Letter 1]

27 Jan 2021

Dear Editor,

Thank you for considering our manuscript for publication in PLOS ONE and for the opportunity to submit a revised version. We are glad that you consider our responses to the reviewers sufficient. We appreciate the constructive criticisms you have raised regarding our figures and statistical tests, and have made appropriate changes in response as detailed point by point below. We hope that this new revision will allow the manuscript to be published in PLOS ONE.

Yours sincerely,

Mélanie Thierry, Nicholas A. Pardikes, Chia-Hua Lue, Owen T. Lewis & Jan Hrček

Additional Editor Comments:

Dear THIERRY, M. and authors

I have gone through the revised manuscript. Comments raised by the reviewers were responded by the authors ok. However, I still have doubts about figures (2, 3 and S7) and statistical tests.

* Thank you for raising those mistakes and oversight. We have corrected the results, figures and the tables as suggested.

Line 227 points out the existence of Tukey's post hoc test, however, I could not find through the text or at the figures any reference to it. Please confirm it and add the information or remove it from the text.

* We corrected the methods, which now reads “Post-hoc multiple comparisons were performed using the emmeans package, and P-values were adjusted using the Tukey method.” [lines 227-228 of the new revised manuscript]. We have added the Post Hoc test results throughout the Result section [lines 348-352 of the new revised manuscript].

Figure 2, 3 and S7 labels ascribe (The small points represent the values from each block and each host-parasitoid pair, the large points represent the grand mean, and the bars represent standard errors of the means). However, I could no find a difference between point sizes. Please confirm it and add the information or remove it from the text.

* Apologies, the small points were lost with the format conversion of the figures. We uploaded revised versions of the figures with both point sizes.

Figure 3(b) label legends species as follow (□: Asobara sp. - D. pseudoananassae, ˄: Asobara sp. - D. birchii, +: Asobara sp. - D. sulfurigaster, × Leptopilina sp. - D. pseudoananassae, ◊: Leptopilina sp. - D. birchii, ˅: Leptopilina sp. - D. sulfurigaster). However, figure did not shows it. Please add it to the figure.

* This was the same problem than for the small points raised previously. The revised version of Figure 3 now shows those symbols.

Tables 2 and S5 labels are quite similar (difference consist only "...for the whole dataset". Please explain the difference between them - by means of the whole dataset - in the S5 label so readers could follow the authors' idea easily. 

* We have added to the Table S5 label “(without any deleted observations due to D. sulfurigaster)”.

Further and regarding to F-values legend [F-values are presented with the significance of the effect: (***) P < 0.001, (**) P < 0.01, (*) P < 0.05, (.) P < 0.1, (ns) P > 0.05"] - I understand that p-values greater than 0.05 are non-significative, so "(.) P < 0.1" does not make sense to me, and I am asking to avoid it, even though R model summary gives it. Then change it at the tables (.) to (ns).

* We changed the P-value significance as suggested.

---

## [Editor Report · Decision Letter 2]

29 Jan 2021

Experimental warming influences species abundances in a Drosophila host community through direct effects on species performance rather than altered competition and parasitism

PONE-D-20-39886R2

Dear Dr. THIERRY,

We’re pleased to inform you that your manuscript has been judged scientifically suitable for publication and will be formally accepted for publication once it meets all outstanding technical requirements.

Kind regards,

Lucas D. B. Faria

Academic Editor

PLOS ONE

Additional Editor Comments (optional):

I have gone through the revised manuscript (R2) and I am satisfied with the changes. So I am pleased to say that the manuscript is accepted to be published in Plos One Journal.

Congratulations and kind regards
---

## [Editor Report · Acceptance letter]

1 Feb 2021

PONE-D-20-39886R2 

Experimental warming influences species abundances in a *Drosophila* host community through direct effects on species performance rather than altered competition and parasitism 

Dear Dr. Thierry:

I'm pleased to inform you that your manuscript has been deemed suitable for publication in PLOS ONE. Congratulations! Your manuscript is now with our production department. 

Kind regards, 

on behalf of

Dr. Lucas D. B. Faria 

Academic Editor

PLOS ONE